# Residual Cough and Asthma-like Symptoms Post-COVID-19 in Children

**DOI:** 10.3390/children10061031

**Published:** 2023-06-08

**Authors:** Abdullah Al-Shamrani, Khalid Al-Shamrani, Maram Al-Otaibi, Ayed Alenazi, Hareth Aldosaimani, Zeyad Aldhalaan, Haleimah Alalkami, Abdullah A. Yousef, Sumayyah Kobeisy, Saleh Alharbi

**Affiliations:** 1Department of Pediatrics, Prince Sultan Military Medical City, Al Faisal University, P.O. Box 7897, Riyadh 11159, Saudi Arabia; 2College of Medicine, Al Marifah University, P.O. Box 92882, Riyadh 11663, Saudi Arabia; drshamrani@outlook.com; 3Department of Pediatrics, Prince Sultan Military Medical City, P.O. Box 26523, Riyadh 12841, Saudi Arabia; meal-otaibi@psmmc.med.sa; 4Respiratory Division, Department of Pediatrics, Prince Sultan Military Medical City, P.O. Box 7456, Riyadh 13326, Saudi Arabia; ammalonazi@psmmc.med.sa; 5Department of Emergency, Prince Sultan Military Medical City, P.O. Box 282236, Riyadh 11392, Saudi Arabia; haldosaimani@psmmc.med.sa; 6Department of Infectious Disease, Prince Sultan Military Medical City, P.O. Box 106383, Riyadh 11666, Saudi Arabia; dr.zeyadaa@hotmail.com; 7Department of Pediatrics, Abha Maternity & Children Hospital, P.O. Box 62521, Abha 3613, Saudi Arabia; haalalkami@moh.gov.sa; 8Department of Pediatrics, King Fahd Hospital of the University, P.O. Box 2208, Al-Khobar 31952, Saudi Arabia; aaayousef@iau.edu.sa; 9College of Medicine, Imam Abdulrahman Bin Faisal University, P.O. Box 1982, Dammam 34212, Saudi Arabia; 10Dr. Soliman Fakeeh Hospital, P.O. Box 2537, Jeddah 21461, Saudi Arabia; skobeisy@fakeeh.care (S.K.); saharbi@uqu.edu.sa (S.A.); 11Department of Pediatrics, Umm Al-Qura University, P.O. Box 715, Mecca 24382, Saudi Arabia

**Keywords:** coronavirus, long COVID, COVID-19, asthma, cough

## Abstract

Background: Coronavirus disease 2019 (COVID-19) has rapidly spread worldwide and is characterized by different presentations ranging from asymptomatic to severe pneumonia. COVID-19 affects all age groups, including pediatric patients. We observed numerous children complaining of a cough post-COVID-19, even if it was trivial. The most reported persistent symptoms after recovery from COVID-19 were insomnia, coughing, fatigue, dyspnea, loss of taste and/or smell, and headache. To date, residual cough post-COVID-19 has been reported in pediatrics and adolescents. Method: we conducted a retrospective study, with a self-administered questionnaire by the patient or caregiver, 12 months post-COVID-19-infection. Result: A total of 94.8% of patients were Saudi citizens and were mainly from the southern region of Saudi Arabia (50.0%). Mothers (64.4%) submitted most of the results. The ages were as follows: 6–14 years (51.0%), 3–5 years (32.3%), and younger than 2 years of age (only 16.7%). Females accounted for 41.7% of those studied. Nearly half of the patients (48.5%) had had a previous COVID-19 infection in 2022, with only 2.1% infected in 2019. Only 27/194 (13.9%) patients required hospital admission, and 7 of them (4.2%) required intensive care treatment. A total of 179 (92.2%) patients still reported persistent symptoms 4 weeks post-COVID-19-infection. A cough was reported in 69.8% of patients, followed by cough and wheezing in 12.3%. The cough was described as dry in 78.0% and nocturnal in 54.1%, while 42.5% did not notice any diurnal variation. For those reporting residual cough, 39.3% found that it affected school attendance and daily activities, 31.1% reported associated chest pain, 51.9% associated it with wheezing, and 27.1% associated it with shortness of breath. For 54.4%, the residual cough lasted less than one month, while 31.4% reported a 1–2 month duration. Only 1.0% had a duration of cough of more than 3 months. For cough relief, 28.2% used bronchodilators, 19.9% used cough syrup, 16.6% used a combination of bronchodilators and steroid inhalers, and 1.7% used antibiotics. Surprisingly, 33% attempted herbal remedies for cough relief. Sesame oil was used the most (40.0%), followed by a mixture of olive oil and sesame oil (25.0%), and 21.7% used male frankincense. The majority (78.4%) sought medical advice for their post-infection cough, either from general pediatricians (39.5%) or via specialist pediatric pulmonology consultations (30.9%). A total of 11.0% with a residual cough reported having pets at home, while 27.2% reported secondhand smoke exposure in the household. Before infection with COVID-19, only 32.6% were diagnosed with asthma, while 68.2% reported a diagnosis of atopic skin. Conclusions: There was a high prevalence of residual cough post-COVID-19, extended for a minimum of two months, and the characteristics of the cough were very similar to those of asthmatic patients. There was still a high prevalence of using cough syrup and herbal remedies, especially olive oil, sesame oil, and male frankincense. A residual cough adversely affected school attendance in daily activities, and there was a high prevalence of other siblings in the family being affected. The study showed that a minority of patients were seen by the pulmonologist; luckily, long COVID was rare in our study, and so further studies are highly needed to confirm the association with asthma. More educational programs are highly needed regarding herbal remedies and cough syrup.

## 1. Introduction

A new beta coronavirus was first identified in December 2019 [1]. Coronavirus disease 2019 rapidly spread worldwide and was characterized by different presentations ranging from asymptomatic to severe pneumonia [1]. The first pediatric COVID-19 case was identified in Shenzhen, China, on 20 January 2020, and on 2 March 2020, the first confirmed case of pediatric COVID-19 was discovered in the Kingdom of Saudi Arabia [2]. COVID-19, after the acute stage, has residual symptoms ranging from mild to severe, referred to by some authors as long COVID [3]. The immediate complications of COVID-19 are well defined and are often associated with increased mortality. However, delayed or long-term complications of COVID-19 are increasingly being recognized and are associated with increased morbidity [4]. Children represented 18.0% of all cases (15,475,992 total child COVID-19 cases) according to the American Academy of Pediatrics on 2 March 2023 [5]. To date, the data have shown that COVID-19 in pediatrics and adolescents affected 15.6 million globally, remaining mild to moderate in the majority (80–90%), with low mortality, and can even be asymptomatic [6,7].

The post-acute COVID-19 clinical features in pediatric patients vary widely and may overlap with allergic symptoms [8]. In the acute stage, several systems are affected. The respiratory system is the most affected, where cough, shortness of breath, and chest pain can persist for several months. Other systems are commonly involved with different complaints: myocarditis, irregular heartbeat, loss of smell and taste, fatigue, insomnia, headache, and encephalitis. However, the severity of acute infection is much lower in children [5,8,9]. At least two long-term consequences that occur following severe acute respiratory syndrome coronavirus 2 (SARS-CoV-2) infection in children are multisystem inflammatory syndrome (MIS-C) and long COVID [10].

Multisystem inflammatory syndrome in children (MIS-C) is a unique medical phenomenon that occurs in children 2 to 6 weeks after infection; it develops in less than 0.1% of children with COVID-19 (median age 8.6 years) and requires intensive care support in 68% of cases [2,11,12,13,14,15,16,17,18,19]. One of the first reports on long-term COVID-19 in children was a case report of five Swedish children (median age: 12 years) who had symptoms lasting 6–8 months after acute respiratory infection symptoms [20]. Sandra Lopez-Leon et al. reported that the prevalence of long-term COVID-19 was 25.24%, and the most prevalent clinical manifestations were mood symptoms (16.50%), fatigue (9.66%), and sleep disorders (8.42%) [10].

The signs and symptoms of long COVID in children have already been described. Moreover, in childhood, the main symptoms concern the neuropsychiatric sphere (insomnia, 18%; fatigue, 11%; and loss of concentration, 10%) [21,22,23,24]. The study aimed to evaluate the prevalence of cough, duration, character, and other respiratory symptoms post-acute-COVID-19-infection among pediatric patients and to further assess the prognostic factors post-COVID-19.

## 2. Method

This retrospective study was conducted in four tertiary centers in the Kingdom of Saudi Arabia (Prince Sultan Military Medical City (PSMMC) in Riyadh, Imam Abdulrahman Bin Faisal University in Dammam, and Dr. Suleiman Fakeeh Hospital and Maternity and Children Hospital in Abha, Saudi Arabia) among children with a physician-confirmed diagnosis of previous COVID-19.

The sample size was calculated using Epitools (https://epitools.ausvet.com.au/oneproportion, accessed on 29 June 2022). A sample of 225 children with previous COVID-19 was the minimum sample to have 80% power to ascertain a prevalence of adverse effects, with a 95% confidence interval and a precision of 5%. It was presented and evaluated by four experts in the field, and a pilot sample of 30 responses was used to assess the validity and reliability of the tool. An electronic questionnaire was used for most patients through social media, e.g., WhatsApp or Twitter, and the remaining patients were evaluated by directly answering a Google questionnaire in the outpatient clinic; the focus was on children with persistent or intermittent cough post-COVID-19-infection. The statistical analysis was performed using SPSS for Windows, version 21.0 (SPSS Inc., Chicago, IL, USA). The data are reported as frequencies and proportions. A non-parametric test was employed for variables with non-normal distribution. Independent sample t-tests were used to compare data between participants of different ages. Chi-square tests were used to compare categorical groups. *p* < 0.05 with a 95% confidence interval was considered to be statistically significant.

The inclusion criteria were as follows: pediatric patients (aged less than 14 years), history of recent COVID-19 infection, and persistent respiratory symptoms for more than 4 weeks (including cough, wheezing, shortness of breath, and stridor). The exclusion criteria were as follows: any patient on medication for a known case of asthma or any patient with comorbidities (congenital heart disease, bronchopulmonary dysplasia, gastroesophageal disease, or metabolic or central nervous system disorder). The definition of COVID-19 infection was any pediatric patient less than 14 years old with a positive COVID-19 polymerase chain reaction.

Beyond the acute phase, long COVID is usually used as a term to describe the persistence or recurrence of health symptoms beyond the acute phase of infection, considered to extend to four weeks [25,26,27,28,29].

Clinical Excellence defined post-COVID-19 syndrome for adults as follows: signs and symptoms that develop during or after an infection consistent with COVID-19, which continue for more than 12 weeks and are not explained by an alternative diagnosis [30]. The World Health Organization definition is as follows: Symptoms occurring at least three months after probable or confirmed SARS-CoV-2 infection. Symptoms must last for at least two months and cannot be explained by an alternative diagnosis [31]. The exact timing, which ranges from 4 to 12 weeks after acute infection, is still an area of debate, although the majority of definitions use 12 weeks as the cut-off point for the definition of long COVID [32,33,34,35]. In our study, we utilized the Robert Koch Institute (RKI) definition, as we believe that COVID-19 in pediatric patients is less severe than that reported in adults, which is defined as a longer-term health impairment following a SARS-CoV-2 infection which is present beyond the acute phase of the sickness of 4 weeks [16,34,36].

## 3. Results

This study included 194/225 participants, who responded via survey, with a history of COVID-19 infection. Parents/guardians answered questions regarding their respective minors, whereby 94.8% of whom were Saudi citizens and were mainly from the southern region of Saudi Arabia (50.0%) (Table 1).

Mothers (64.4%) submitted most of the results. The study subjects ranged in age, with the most common age group being those between 6 and 14 years (51.0%), followed by those between 3 and 5 years (32.3%). Only 16.7% were younger than two years of age. Females accounted for 41.7% of those studied.

The COVID-19 symptoms for 22.7% were cough and runny nose, followed by cough alone in 16.0%, and 15.5% reported cough, wheezing, and shortness of breath. A total of 179/194 (92.2%) patients still reported persistent symptoms post-COVID-19-infection. A cough was reported in 69.8% of patients, followed by cough and wheezing in 12.3%.

Most children had previous COVID-19 infection in 2022 (48.5%), with only 2.1% infected in 2019, which is doubtful, as the first reported case was in March 2020 and parents could not recall the exact time of infection. Only 27/194 (13.9%) patients required hospital admission for further management, and 7 (4.2%) required intensive care treatment (Table 2). The cough was described as dry in 78.0% and nocturnal in 54.1%, while 42.5% did not notice any diurnal variation. For those reporting a residual cough, who required a doctor visit, 39.3% found that it affected school attendance and daily activities. In addition, 31.1% reported associated chest pain, 51.9% associated their residual cough with wheezing, and 27.1% associated their residual cough with shortness of breath. For 29.2%, the cough was triggered by cold exposure, and 17.8% attributed cold and incense as triggers. For 53.8%, the residual cough lasted less than one month, while 30.1% reported a 1–2 month duration. Only 1.1% had a cough duration of more than 4 months.

For cough relief, 28.2% used bronchodilators, 19.9% used cough syrup, and 16.6% used a combination of bronchodilators and steroid inhalers. Antibiotics were only used by 1.7% of patients. The majority (78.4%) sought medical advice for their post-infection cough; 39.5% saw general pediatricians, while 30.9% sought out specialist pediatric pulmonology consultations.

A total of 93.7% reported that other family members were infected with COVID-19, with 83.2% reporting that more than two family members were affected. Surprisingly, only 33% attempted herbal remedies for cough relief. Sesame oil was used the most (40.0%), followed by a mixture of olive oil and sesame oil (25.0%), and 21.7% used male frankincense. Only 11.0% with a residual cough reported having pets at home. In contrast, 27.2% reported secondhand smoke exposure in the household. Before infection with COVID-19, only 32.6% were diagnosed with asthma at least once; however, those on regular preventive medications were excluded to minimize the potential biases related to asthma, and 68.2% reported a diagnosis of atopy.

## 4. Discussion

The evidence on post-acute COVID-19 conditions and long-term outcomes in pediatric patients is still limited to small studies, with a lack of a control group [10,15,16], and the clinical features in pediatric patients vary widely [11]. The nature of the long-term outcome of COVID-19 is unpredictable, since persistent symptoms can be continuous or relapsing and remitting, making it difficult to establish a diagnosis [37]. Cough (4%) and fatigue (2%) are residual post-COVID-19-infection symptoms [38]. The current data are not specific enough to separate long COVID from other COVID-19 mimickers, such as viral infection [39]. During the COVID-19 pandemic, the application of personal protective measures and public health measures resulted in a marked reduction in the spread not only of coronavirus but also of the circulation of other respiratory viruses, and several studies were published and proved this [40,41,42,43].

The duration of COVID-19 was markedly variable due to the lack of standardization of the definition and severity of COVID-19 in the literature, which was reported to be as low as 4–4.6% by Radtke et al. [44] and Miller et al. [45] and as high as 100% by Brackel et al. [46] and Ashkenazi Hoffnung et al. [47].

In England and Wales, the VirusWatch study collected data from 4678 children. Symptoms lasting for more than four weeks were reported by 4.6%, compared to by 1.7% in the control group [45]. Meta-analyses of the prevalence of long COVID-19 showed that the prevalence of long COVID in children and adolescents, as defined by the presence of one or more symptoms for more than 4 weeks following COVID-19 infection, was 25.24% [10]. Daniela Say et al. reported a cough, which may persist up to 8 weeks from the onset of symptoms, in 4% of patients [39]. Danilo Buonsenso et al., in a cross-sectional study, reported that 53% of pediatric patients had at least one persistent symptom for 4 months or more after a diagnosis of acute SARS-CoV-2 infection [24]. A similar study from Latvia reported that 22% of patients noted three or more persistent symptoms 2 months post-COVID-19 [8]. In Israel, a prospective, multicenter cohort study in 90 patients with long COVID showed that the median number of chronic symptoms was four, and, in almost 60% of patients, symptoms were associated with functional impairment one to seven months after illness onset [47]. Karel Kostev et al. reported a low prevalence of post-COVID-19 in children (1.7%) one month after acute illness, which was higher in adolescents aged 13–17 years old compared to children less than 5 years old (38.4: 23%). Allergic rhinitis (12.3%), somatoform disorders (7.9%), and anxiety disorders (5.3%) were also significantly associated with a post-COVID-19 condition [48]. Similarly to our study, the most common age group was between 6 and 14 years (51.0%), followed by 3–5 years (32.3%). Only 16.7% were younger than two years of age. In a study by Liene Smane et al., among 92 children, 51% reported at least one symptom out of tiredness, loss of taste and/or smell, and headaches by 1–3 months, with the majority being 10–18 years of age [8]. Ieva Roge et al. reported at least one persistent symptom in 236 children (53%). The most common complaints were persistent fatigue (25.2%); cognitive sequelae, such as irritability (24.3%), mood changes (23.3%), headaches (16.9%), rhinorrhea (16.1%), and coughing (14.4%); and anosmia/dysgeusia (12.3%). In addition, 44.5% of COVID-19 patients had persistent symptoms after the 12-week cutoff point, with irritability (27.6%), mood changes (26.7%), and fatigue (19.2%) being the most reported symptoms [40]. In infants and preschoolers, upper respiratory tract symptoms such as cough and rhinorrhea were the most commonly reported, followed by diarrhea and nocturnal sweating. In older children, fatigue, cognitive disturbances, and neurological sequelae were the most prevalent signs and symptoms [22,40]. Anosmia, especially in the pediatric field, was often underdiagnosed [49,50].

Compared to our study, which is similar to the literature, residual cough after 4 weeks lasted less than one month for 54.4%, while 31.4% reported a duration of 1–2 months, 13.2% reported more than 2 months, and only 1% had a long COVID or cough duration of more than 3 months [15]. In most studies, symptoms did not persist beyond 12 weeks [44,50,51,52].

Long COVID is well known in pediatric patients (the prevalence was 25.4%, as reported by Sandra Lopez-Leon et al.), and symptoms are resolved within 1–5 months [10,16]. Earlier recovery within 2 weeks to 3 months was previously described in pediatric studies [50,51,52,53,54]. Slower recovery was reported by Balderas et al., where 9.3% at 4 months and 2.3% at 6 months were still coughing [55]. The reported mortality was 0.4% [15,55], and children and adolescents were more likely to be subsequently diagnosed with post-COVID-19 conditions than children aged less than 5 years [8,35].

In Saudi Arabia, A. AlRadini et al. reported that 70% of patients were symptomatic with five or fewer symptoms. In a multicenter, retrospective cross-sectional study of SARS-CoV-2 infection involving 314,821 patients, late symptoms lasting for more than 4 weeks included loss of smell, loss of taste, fatigue, shortness of breath, and cough (52.4%, 31.1%, 11.5%, 10.2%, and 8.9% of patients, respectively) [55]. In our study, the presenting COVID-19 symptoms for 22.7% were cough and runny nose, followed by cough alone in 16.0%. A total of 15.5% reported cough, wheezing, and shortness of breath, which is a little different from AlRadini’s results. A total of 179/194 (92.2%) patients still reported late symptoms of post-COVID-19-infection, higher than what was reported by AlRadinit et al., with a predominance of respiratory symptoms, including cough alone in 69.8%, followed by cough with wheezing in 12.3%.

The cough was described as dry in 78.0% and as nocturnal in 54.1%, while 42.5% did not notice any diurnal variation. For those reporting residual cough, 39.3% found that it affected school attendance and daily activities, 31.1% associated it with chest pain, 51.9% associated it with wheezing, and 27.1% associated it with shortness of breath. For 29.2%, the cough was triggered by cold exposure, and 17.8% attributed cold and incense as triggers; such symptoms are very suggestive of being asthma-like, since COVID-19 is another virus that can trigger asthma. Asthma is common in Saudi Arabia, with a prevalence of 23%, per Al Frayh AR et al. [56]. Allergic diseases and older age are the main risk factors for persistent symptoms [15]. In our study, before infection with COVID-19, only 32.6% were diagnosed with asthma, and 68.2% reported a diagnosis of atopy. Most asthma exacerbations are caused by viral respiratory infections such as rhinovirus, respiratory syncytial, coronaviruses, and influenza [57,58]. The severity is worse in asthmatic patients than in non-asthmatic patients, and the pathophysiology is still obscure and may involve humoral and cell-mediated immunity arm defects, while adequate asthma control is associated with a significant decrease in episodes of exacerbation [58]. AlRadinia et al. reported that children with a history of allergic diseases (i.e., asthma, allergic rhinitis, eczema, and food allergy) are at a significantly increased risk for post-COVID-19 conditions, compared with their counterparts without a history of allergic diseases [55].

In our study, for cough relief, 28.2% used bronchodilators, 19.9% used cough syrup, and 16.6% used a combination of bronchodilators and steroid inhalers. Antibiotics were only used in 1.7% of patients. The majority (78.4%) sought medical advice for their post-infection cough. Surprisingly, only 33% attempted herbal remedies for cough relief. Sesame oil was used the most (40.0%), followed by a mixture of olive oil and sesame oil (25.0%), and 21.7% used male frankincense. However, we did not focus on non-respiratory symptoms such as fatigue, sleep problems, and sensory changes [22].

The reported risk could increase post-COVID-19 with increasing age [8,15], while self-reported symptoms can be significantly biased in younger children, as they cannot adequately express their functional status or emotional relevance post-COVID-19 [10]. Only 27/194 patients required hospital admission for further management, and 7 of them required intensive care treatment. Children who were hospitalized for 48 h and were presenting more than four symptoms were more liable for long COVID [59,60].

In our study, we found that long COVID was more common in males and patients with at least one sibling with similar symptoms. In contrast, we did not find that having pets or smoking increased coughing in patients with long COVID. This is different from what was reported in the reported risk factors for long COVID in adults, including female sex, middle age, white ethnicity, and comorbidities, especially asthma [24,38,57]. There are less data on long COVID in children and adolescents [61,62,63,64]. General pediatricians were consulted by 39.5%, while specialist pediatric pulmonology consultations were sought out by 30.9%.

The main limitations of our study are the small sample size, a shorter duration of long COVID definition, and a parental-dependent questionnaire; furthermore, we did not address non-respiratory manifestation. We tried to reach a larger sample, but the disease has declined globally due to massive vaccinations and few new cases. This manuscript focused on respiratory symptoms rather than long COVID. Another limitation was that vaccination was not explored.

## 5. Conclusions

There was a high prevalence of residual cough post-COVID-19, luckily being resolved within two months, and the characteristics of the cough were very similar to those of asthmatic patients. There was still a high prevalence of cough syrup and herbal remedies being used, especially olive oil, sesame oil, and male frankincense. A residual cough adversely affected school attendance in daily activities, and there was a high prevalence of siblings often being affected. This study showed that long COVID is rare, and a minority of the patients were seen by a pulmonologist. Appropriate asthma therapies could improve the outcome; so, further studies are needed to confirm the association with asthma. The accurate determination of the risk of long COVID is still debated and is a relevant area of research.

## Figures and Tables

**Table 1 children-10-01031-t001:** Patient demographics.

Characteristic	*N* (%)
Sex:	
Male	112 (58.3)
Female	80 (41.7)
Age group:	
<2 years	32 (16.7)
3–5 years	62 (32.3)
6–14 years	98 (51.0)
Nationality:	
Saudi	182 (94.8)
Non-Saudi	10 (5.2)
Region of Saudi Arabia:	
Central	53 (27.9)
Eastern	15 (7.9)
Northern	1 (0.5)
Southern	95 (50.0)
Western	26 (13.7)

**Table 2 children-10-01031-t002:** Questionnaire answers.

Question	*N* (%)
What year did your child contract COVID-19?	
2019	4 (2.1)
2020	36 (18.6)
2021	60 (30.9)
2022	94 (48.5)
At the time of COVID-19 diagnosis, was your child admitted to the hospital?	
Yes	27 (13.9)
No	167 (86.1)
If admitted, was ICU admission required?	
Yes	7 (4.2)
No	159 (95.8)
If your child still has a cough after COVID-19, how is it best described?	
Dry	142 (78.0)
Wet	40 (22.0)
What time of day does your child experience the cough?	
During the whole day and night	7 (42.5)
Only during the daytime	6 (3.3)
Only at night	98 (54.1)
Is your child’s daily activity or school attendance affected by the cough?	
Yes	72 (39.3)
No	111 (60.7)
Is the cough associated with any of the following:	
Chest pain?	57 (31.1)
Wheezing?	95 (51.9)
Shortness of breath?	49 (27.1)
Did your child require a doctor’s visit due to the cough?	
Yes	145 (78.4)
No	40 (21.6)
Did your child use herbal remedies for the cough?	
Yes	62 (33.0)
No	126 (67.0)
Before being infected with COVID-19, was your child asthmatic?	
Yes	62 (32.6)
No	128 (67.4)

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
