# Peer review of "Residual Cough and Asthma-like Symptoms Post-COVID-19 in Children"

_children, 2023, doi:10.3390/children10061031_

Round 1

Reviewer 1 Report

The coronavirus disease 2019 (COVID-19) pandemic, after the acute stage, has residual symptoms ranging from mild to severe, and referred to by some authors as long COVID-19. In this study the authors aimed to evaluate the prevalence of cough and respiratory symptoms post-acute  COVID-19 infections among pediatric patients and to further assess the prognostic factors  post-COVID-19.

The manuscript is quite interesting. The article is overall well written, but it would benefit from some revisions.

·         The introduction section is adequately written, providing all necessary information to the readers;

·          Line 109An electronic questionnaire was used for most patients, and the other patients were evaluated with a direct questionnaire or through the clinic How was the questionnaire realized? Please explain the methodology and the administration tools; It is fundamental you specify this step in order to avoid biases on results;

·         Table 2 Before being infected with COVID-19, was your child asthmatic? Yes No 62 (32.6) 128 (67.4). 32.6% of parents declared their sons were asthmatic before the COVID-19 infection. In this patients it is difficult to assess the nature of respiratory symptoms declared since they could be due to a previous asthmatic condition. This could be a bias on your results. Please make appropriate explanations;

·         Line 174-175 The current data  are not specific enough to separate long COVID from other COVID-19 mimickers, such as viral infection [31]. Please mention in discussion what was the impact of COVID-19 on other respiratory infections during the pandemic. Please cite and make reference to Di Sarno L, Curatola A, Conti G, Covino M, Bertolaso C, Chiaretti A, Gatto A. The effects of COVID-19 outbreak on pediatric emergency department admissions for acute wheezing. Pediatr Pulmonol. 2022 May;57(5):1167-1172. doi: 10.1002/ppul.25858. Epub 2022 Feb 22. PMID: 35170263; PMCID: PMC9088495.

·         The authors should not repeat in discussion results without any comments.

·         The main objective of the study is clear but the analysis of results should be more specific and discussed with appropriate updated references in order to support your speculation.

A careful English language revision of every section is suggested.

Author Response

THANKS, GREAT COMMENT

 VALUABLE FEEDBACK 

SHAMRANI

Reviewer 2 Report

A statistical description of the associated cases of diagnosed bronchial asthma is also useful, as well as the description of the clinical manifestations, comparing the categories of patients affected by Covid 19 and those with associated bronchial asthma.

We also consider the description of the period and the number of times the questionnaire was applied to be of interest, we note that the data collection is based more on the description of the symptoms by the parents and is not based on a clinical examination. This can affect the veracity of the described data because the subjectivity of the parents intervenes, they will tend to aggravate the children's symptoms.

Author Response

Thanks a lot of the valuable feed back 

 appreciate the encouraging comment 

Reviewer 3 Report

Well written work on cough and asthma-like symptoms in post covid. Reduce the abstract, it is too long and wordy. In the introductory part you could enrich this part by saying that there have been many symptoms, especially in paediatrics, which overlap with allergic symptoms. Mostly the symptom of anosmia especially in the pediatric field was often underdiagnosed.You can refer to these works below:

- Brindisi G, De Vittori V, De Castro G, Duse M, Zicari AM. Pills to think about in allergic rhinitis children during COVID-19 era. Acta Paediatr. 2020 Oct;109(10):2149-2150. doi: 10.1111/apa.15462. Epub 2020 Jul 19. PMID: 32627237; PMCID: PMC7361544.

-Brindisi G, De Vittori V, De Nola R, Pignataro E, Anania C, De Castro G, Cinicola B, Gori A, Cicinelli E, Zicari AM. Updates on Children with Allergic Rhinitis and Asthma during the COVID-19 Outbreak. J Clin Med. 2021 May 24;10(11):2278. doi: 10.3390/jcm10112278. PMID: 34073986; PMCID: PMC8197398.

The statistic part is very simple,  only descriptive. The sample size and the discussion of the results achieved are good. Minor english revision required.

Only minor revision required.

Author Response

 great and valuable constructive feedback 

 many thanks 

dr shamrani

Reviewer 4 Report

The manuscript needs major revisions.

line 28-29: the verb is missing in the sentence.
line 33 'A total of 179 (92.2%) patients still reported persistent symptoms post-COVID-19 infection': how long after the COVID-19 infection? 

line 39-40: 'For 53.8%, the residual cough lasted less than one month, while 30.1% reported a 1–2-month duration. Only 1.1% had a duration of cough of more than 4 months': compared to acute infection, healing? The time reference is missing.

line 45: capital letters after dots
line 55-57: not clear, explain better
line 60-63: use the verb in the past tense
line 66: The abbreviation COVID-19 should be used the first time Coronavirus disease 2019 is mentioned

line 72-74: unclear sentence, rephrase
line 80-82: 'At least two long-term consequences that occur following severe acute respiratory syndrome coronavirus 2 (SARS-CoV-2) infection in children: multisystem inflammatory syndrome (MIS-C) and long-term COVID-19': the verb is missing in the sentence. What is meant by long-term consequences? MIS-C can arise 2 to 6 weeks after acute infection, while long COVID refers to symptoms that persist 3 months after infection (WHO definition). Perhaps it would be better to talk about medium and long-term consequences
line 82: replace long term COVID with long COVID
line 93-94: 'The signs and symptoms were well described in adults, including fever, fatigue, and dry cough and similar symptoms were reported in children with COVID-19'. It is a repetition, the symptoms of long COVID in children have already been described. Moreover, in childhood the main symptoms concern the neuropsychiatric sphere.
line: 94: explain the study's purpose more clearly in light of the persistent respiratory issues documented in children.
line 118-120 'The inclusion criteria were as follows: pediatric patients (age less than 14 years), history of acute COVID-19 infections, and respiratory symptoms post-acute COVID infection (12 months), including cough, wheezing, SOB, stridor and unknown medical illness or atopy.' : SOB is the acronym, write what it refers to. What does '12 months' mean? Children who have had COVID-19 in the last year? Specify better. Furthermore, it must be indicated whether the symptoms appeared/persisted after weeks and how many weeks after the infection, otherwise we cannot speak of long COVID. 'Unknown medical illness or atopy?' What does it mean?
line 126-132: See also WHO guidelines (Comparison of definitions of long COVID--->Scharf, R.E.; Anaya, J.-M. Post-COVID Syndrome in Adults An Overview. Viruses 2023, 15, 675. https://doi.org/10.3390/v15030675)
line 143-144: 'A total of 179/194 (92.2%) patients still reported persistent symptoms post-COVID-19 infection': how long after the COVID-19 infection? How many of them can actually be included in the definition of long COVID?
line 167: asthmatic*
line 146-147: 'Most children had previous COVID-19 infection in 2022 (48.5%), with only 2.1% infected in 2019': how they were diagnosed COVID-19 in 2019 if you previously said that 'on the second of March 2020, the first confirmed case of pediatric COVID-19 was discovered in the Kingdom of Saudi 65 Arabia' (line 64-65).
line 216: 'Long COVID is rare, and symptoms resolve within 1–5 months: Long COVID is not rare. The prevalence in childhood is about 25%.
line 218: Compared to our study (53.8%): what does it mean?
line 258-259: 'In our study, we found that long COVID-19 was more common in males and patients with at least one sibling with similar symptoms': in order to be able to say that the children had long COVID, it is necessary to specify whether the symptoms persisted after weeks and how many weeks after the acute infection.
line 272-273: 'The manuscript focused on respiratory symptoms rather than the long COVID-19': Long COVID includes also respiratory symptoms. This sentence is a contradiction with everything previously written because the whole manuscript talks about long COVID, it must be specified whether these are respiratory symptoms that are part of the long COVID or residual symptoms that are resolved in a short time FROM THE ACUTE INFECTION. In this case they would not be long COVID symptoms, but residual symptoms. Long COVID is a specific diagnosis.
The discussion should be rewritten. The results should not be repeated but should be discussed in the context of other studies in the literature, highlighting the differences for example 'a total of 179 (92.2%) patients still reported persistent symptoms post-COVID-19 infection. Cough was reported in 69.8% of patients, followed by cough and wheezing in 12.3%'. It is a very high prevalence, why was this trend observed compared to other studies? Were patients with respiratory symptoms also followed up? Have they performed any more in-depth tests such as spirometry or lung ultrasound?   In general, the manuscript needs a major revision. The purpose is not clear because the methods used are not well understood. Did the children eventually have long COVID or not?     

Author Response

no much words to thanks this reviewer for his/her amazing feedback 

hope that we satisfy him/her

 certificate of editing available 

 dr shamrani

Round 2

Reviewer 2 Report

I noted your response.

Reviewer 4 Report

 In this form, the manuscript can be accepted.